# OpenReview forum: "Too Easily Fooled? Prompt Injection Breaks LLMs on Frustratingly Simple Multiple-Choice Questions"
_ICLR.cc/2026/Conference — ICLR 2026 Conference Withdrawn Submission_

### Official Review · Reviewer_r4Wf · 2025-10-14

**Soundness:** 2
**Presentation:** 3
**Contribution:** 2
**Rating:** 2
**Confidence:** 5

**Summary:**

This paper investigates the vulnerability of LLMs to prompt injection when used in an "LLM-as-a-judge" capacity for grading. The authors create PDF documents with simple arithmetic problems and embed malicious instructions, using either visible black text or invisible white text, to command the model to output a specific, incorrect answer. The results show that many LLMs are indeed fooled by these prompts, particularly the visible ones, while models with a "thinking" mode exhibit greater robustness against the "hidden" white-text attacks.

**Strengths:**

1. The paper studies the reliability of LLM-as-a-judge applications under prompt injection, which is a timely area as AI integration becomes more common.
2. The paper is generally well presented and easy to understand.

**Weaknesses:**

1. The core discovery is LLMs are "fooled" with injected prompts. However, essentially it's an instruction-following model following instructions (injected prompt) found in the input text. The finding is expected and not surprising. The experiments, for example black-text and white-text prompts in pdf, do not provide much insight. The scientific or technical contribution is quite limited.
2. Table 1 contains objectively wrong parameter counts for GPT-4o, o3, and DeepSeek.
3. The authors mentioned the potential impact of prompt injection on peer review (without experiments). It's also a known issue and I believe conferences are aware of this issue and even have rules on it. I'm not sure what the claim is here.

**Questions:**

NA

---

> ### Author Response · Authors · 2025-11-23
>
> Thank you for your thoughtful feedback. Your comments are very helpful and much appreciated. We will address these in the next version.

---

### Official Review · Reviewer_r6G3 · 2025-10-15

**Soundness:** 2
**Presentation:** 1
**Contribution:** 1
**Rating:** 2
**Confidence:** 5

**Summary:**

This paper studies the vulnerability of LLM-as-a-judge to adversarial prompt injection. The work presents a preliminary analysis, showing that the evaluation process of LLMs can be easily twisted by certain prompts, resulting in biased judgments.

**Strengths:**

The paper discusses the important issue of fairness and robustness of LLMs when used as evaluators. The authors' initial exploration is promising. By successfully applying different prompt injection attacks on powerful LLMs, the paper effectively demonstrates the existence of these vulnerabilities.

**Weaknesses:**

Despite the promising direction, the paper suffers from several major weaknesses in its current form:

1. The main conclusion of the paper appears to be trivial and somewhat obvious. The vulnerability of LLMs to injection attacks has already been well-established in a large body of prior work. This paper's analysis, while confirmatory, does little more than reiterate this known phenomenon. Consequently, the primary research question 1 posed by the authors is not a true research question, as its answer is largely self-evident from existing literature.

2. The paper is underdeveloped in several key aspects, including its motivation, methodology (which is not clearly defined), experimental design, and depth of analysis. The work currently reads more like a preliminary study. A more impactful contribution would involve exploring how to mitigate these vulnerabilities. For example, the authors could investigate methods to enhance the LLM's inherent robustness against such attacks to ensure reliable evaluation outcomes.

3. Overall, the current manuscript resembles a technical report or a simple experimental analysis rather than a rigorous academic paper. The contribution is not substantial enough for a publication at this venue.

**Questions:**

Please refer to the Weaknesses.

---

> ### Author Response · Authors · 2025-11-23
>
> We appreciate your constructive suggestions and careful review. Thank you for helping us improve our manuscript. We will incorporate these points in the next version.

---

### Official Review · Reviewer_MD8p · 2025-10-28

**Soundness:** 1
**Presentation:** 3
**Contribution:** 1
**Rating:** 0
**Confidence:** 4

**Summary:**

The authors propose a jailbreaking method for LLMs by injecting “hidden prompts” into PDF documents. The authors show that by changing the color of text embedded in a PDF document nefarious actors could inject information to an LLM that is invisible to people. They show that this is a concern for state of the art LLMs. The authors attempt to demonstrate the effectiveness of this attack with a limited number of examples in the form of true or false and multiple choice questions. They also investigate the impact of "thinking" on the models susceptibility of the attack.

**Strengths:**

The paper is overall well written and straightforward to read. The authors also aim to contribute to an important area, LLM security.

**Weaknesses:**

- **This work seems to lack novelty.** The authors overlook a key related work “Invisible Prompts, Visible Threats: Malicious Font Injection in External Resources for Large Language Models” published in EMNLP Findings 2025 which investigates very similar “font injection” attacks. The setting they investigate is more general and their experimental investigation is more extensive.
- **The experimental investigation is very limited.** It seems like the authors only do an experimental evaluation on 4 total problems. Tables 2, 3, and 4 only cover the results for single problems. The rigor of these experiments could be strengthened by scaling up the investigation to more problems.
- In my opinion, **the implications and main takeaways of this study are quite limited.**

**Questions:**

I don't have any substantial questions.

---

> ### Author Response · Authors · 2025-11-23
>
> Thank you for your detailed comments. They provide clear guidance and will strengthen our work. We will include these changes in the next version.

---

### Official Review · Reviewer_v7xL · 2025-11-02

**Soundness:** 2
**Presentation:** 2
**Contribution:** 2
**Rating:** 2
**Confidence:** 4

**Summary:**

This paper evaluates whether hidden textual prompts embedded in PDF files (e.g., white-colored text or invisible LaTeX instructions) can manipulate LLMs’ answers to trivial arithmetic questions.
The authors test six models (GPT-4o, GPT-o3, Gemini 2.5 Flash/Pro, DeepSeek-V3/R1) under no prompt, black prompt, and white prompt conditions.
They find that even top-tier LLMs can be misled, particularly by visible (black) injections, while “thinking-enabled” models show better robustness.

**Strengths:**

**1) Clear, reproducible setup:** The authors define a simple, interpretable pipeline (Eq. 1) for prompt injection in PDFs using LaTeX color control.

**2) Novel variant of a known problem** Prior works (e.g., Liu et al., 2024c; Guo et al., 2024; Raina et al., 2024) study prompt injection generally, but few explore the PDF-hidden variant. The work highlights this under-studied vector.

**3) Empirical value.** Confirms that even simple visual-level attacks can bypass superficial safety filters in reasoning-oriented LLMs, relevant to "LLM-as-a-judge" systems.

**Weaknesses:**

**1) Trivial methodology, no insight beyond anecdote:** This paper's contribution is essentially a reproduction with simpler math tasks. The authors never quantify why certain models succumb, i.e., there is no causal insight, just observed failure.

**2) Excessive space on prompt instantiation, minimal analysis:** Over two pages are devoted to LaTeX examples of "black", "white", and "no" prompt injections. This is implementation detail; the space could instead show token-level model traces or why the white prompt affects GPT-4o but not DeepSeek-V3.

**3) Overclaiming in title and abstract:** "Breaks LLMs on Frustratingly Simple Questions" is misleading: the white-text attack consistently fails against several tested models (e.g., DeepSeek-V3). At best, the results show partial vulnerability, not systemic failure.

**4) No discussion or evaluation of defenses:** Appendix C adds a one-line "defensive prompt" but provides no systematic mitigation framework.

**5) Missing quantitative metrics:** The analysis is purely categorical ("correct" vs "incorrect"), lacking statistics such as attack success rate = #misled / #total.

**Questions:**

1) Why does the white prompt succeed only on GPT-4o? Is it due to OCR parsing vs PDF text extraction?


2) Do thinking models resist because of reasoning steps or just input preprocessing differences?

3) How many total prompts per model were tested? Please report success rate (%) rather than per-instance tables.

---

> ### Author Response · Authors · 2025-11-23
>
> We are grateful for your review and the helpful points you raised. Thank you for your support. We will address all of them in the next version.

---

### Note · Authors · 2025-11-23

**Comment:**

We would like to sincerely thank all the reviewers for providing insightful comments to improve our work. After careful consideration, we decide to withdraw this paper.

**Withdrawal Confirmation:**

I have read and agree with the venue's withdrawal policy on behalf of myself and my co-authors.